# Mechanisms of Resistance to PI3K Inhibitors in Cancer: Adaptive Responses, Drug Tolerance and Cellular Plasticity

**DOI:** 10.3390/cancers13071538

**Published:** 2021-03-26

**Authors:** Sarah Christine Elisabeth Wright, Natali Vasilevski, Violeta Serra, Jordi Rodon, Pieter Johan Adam Eichhorn

**Affiliations:** 1Faculty of Health Sciences, Curtin Medical School, Curtin University, Bentley 6102, Australia; pieter.eichhorn@curtin.edu.au; 2Curtin Health Innovation Research Institute and Faculty of Health Sciences, Curtin University, Bentley 6102, Australia; 3Vall d’Hebron Institute of Oncology (VHIO), Vall d’Hebron University Hospital, 08035 Barcelona, Spain; vserra@vhio.net; 4MD Anderson Cancer Center, Investigational Cancer Therapeutics Department, Houston, TX 77030, USA; JRodon@mdanderson.org; 5Cancer Science Institute of Singapore, National University of Singapore, Singapore 117599, Singapore; 6Department of Pharmacology, Yong Loo Lin School of Medicine, National University of Singapore, Singapore 117597, Singapore

**Keywords:** PI3K pathway, mechanisms of resistance, PI3K pathway inhibitors

## Abstract

**Simple Summary:**

The phosphoinositide-3-kinase (PI3K) pathway is the most frequently activated pathway in human cancers. Consequently, a number of compounds targeting the various nodes of this pathway have been developed. However, the majority of these compounds have been unsuccessful in patients due to high levels of toxicity, as well as their inability to effectively downregulate the pathway to levels required for tumour responses. This inability to downregulate the pathway is partially mediated by intrinsic adaptive response, also known as compensatory mechanisms or feedback loops, which reactivate the pathway following inhibition; limiting the effectiveness of these compounds. In this review we highlight the mechanisms of action of these adaptive responses and highlight potential combinatorial strategies to delay tumour progression.

**Abstract:**

The phosphatidylinositol-3-kinase (PI3K) pathway plays a central role in the regulation of several signalling cascades which regulate biological processes such as cellular growth, survival, proliferation, motility and angiogenesis. The hyperactivation of this pathway is linked to tumour progression and is one of the most common events in human cancers. Additionally, aberrant activation of the PI3K pathway has been demonstrated to limit the effectiveness of a number of anti-tumour agents paving the way for the development and implementation of PI3K inhibitors in the clinic. However, the overall effectiveness of these compounds has been greatly limited by inadequate target engagement due to reactivation of the pathway by compensatory mechanisms. Herein, we review the common adaptive responses that lead to reactivation of the PI3K pathway, therapy resistance and potential strategies to overcome these mechanisms of resistance. Furthermore, we highlight the potential role in changes in cellular plasticity and PI3K inhibitor resistance.

## 1. Introduction

The phosphatidylinositol-3-kinase (PI3K) pathway is one of the most frequently enhanced oncogenic pathways in human cancers. Since the first discovery of PI3K pathway mutations in solid malignancies in 2004, numerous studies have highlighted the prognostic and therapeutic implications of these mutations. As a result, more than 40 compounds targeting key components of this pathway have been tested in early phase clinical trials. Frustratingly, the clinical development of many of these compounds has not advanced past late phase clinical trials with the efficacy of PI3K inhibitors being primarily limited by their narrow therapeutic window and frequent treatment-related toxicities. Nevertheless, the PI3K inhibitors BYL719 (alpelisib), CAL101 (idelalisib) and BAY 80-6946 (copanlisib), and the mTOR inhibitors RAD001 (everolimus) and CCI-779 (temsirolimus), have been approved for the treatment of various malignancies. However, the outstanding question remains as to why PI3K inhibitors have not yielded the same impressive clinical activity with what is observed in other targeted therapies. In this review, we will discuss some of the known intrinsic adaptive responses which lead to reactivation of the PI3K pathway, drug tolerance or changes in cellular plasticity which together limit responses to PI3K inhibitors in the clinic.

## 2. PI3K Pathway

PI3K belongs to a family of lipid kinases which are responsible for the phosphorylation of the 3′OH group on phosphatidylinositols (PtdIns) [1,2]. PI3Ks are grouped into three classes—I, II or III—based on their physical structure and substrate specificity [3]. Additionally, class I is further subdivided into IA and IB. Class IA PI3Ks are commonly known for the role they play in tumorigenesis in human cancers [4]. Class IA PI3Ks are heterodimers comprised of a p85 regulatory subunit and p110 catalytic subunit. The regulatory subunit can be found in three different isoforms including p85α (encoded by *PIK3R1*), p85β (encoded by *PIK3R2*) and p55γ (encoded by *PIK3R3*) [5,6]. The *PIK3R1* gene also transcribes for two shorter isoforms, p55α and p50α, through alternative transcription-initiation sites. The p85 regulatory isoforms encode for an adaptor-like protein that has two Src-homology 2 (SH2) domains and an inter-SH2 domain that binds constitutively to the catalytic subunit p110, rendering it inactive [2]. Class IA p110 can be found in cells, as one of three isoforms p110α, p110β and p110δ (encoded by *PIK3CA, PIK3CB* and *PIK3CD,* respectively). These three catalytic subunits can associate with any of the five regulatory subunits [7]. Class IB PI3Ks consist of a heterodimer of p110γ catalytic subunit (encoded by *PIK3CG*) and either the regulatory subunit p101 (encoded by *PIK3R5*) or p87, sometimes referred to as p84 (encoded by *PIK3R*6) [3].

PI3Ks are autoactivated by various extracellular stimuli including hormones, cytokines and growth factors which bind to their cognate receptor tyrosine kinases (RTKs) [8,9]. Upon stimulation RTKs are phosphorylated at a conical YXXM motif, which results in the recruitment of the PI3K heterodimer to the plasma membrane and the binding of the p85 SH2 domains to the phosphorylated tyrosine residue [10]. The binding of p85 relieves the inhibition of p110 where p110 is now able to catalyse the generation of phosphatidylinositol 4,5-biphosphate (PIP2) phosphorylation to phosphatidylinositol 3,4,5-trisphosphate (PIP3) [11] (Figure 1).

PIP3 has now been recognised as one of the most important secondary messengers in the cell [12]. The accumulation of PIP3 at the membrane results in the recruitment of Pleckstrin-homology (PH) domain-containing proteins [12]. Approximately 40 mammalian proteins have been identified to date containing this motif including the AKT family of serine–threonine kinases. There are three AKT isoforms (AKT1, AKT2 and AKT3) with each isoform having specific features and distinct roles in cell signalling. All three forms of AKT possess similar structures which allows them to be activated in a similar manner. AKT contains a PH domain that is docked in the N-terminal region to PIP3 [13]. The binding of AKT to PIP3 induces AKT to undergo a conformational change that exposes two amino acids essential for phosphorylation. Furthermore, the binding to PIP3 permits its localisation to another PH domain containing protein kinase, phosphoinositide-dependent protein kinase 1 (PDK1). PDK1 phosphorylates AKT at threonine 308 (T308) in the activation, or T-loop, of the kinase [14]. This phosphorylation mediates the partial activation of AKT kinase activity with a secondary phosphorylation within the C-terminal hydrophobic motif of AKT serine 473 (S473), mediated by the mammalian target of rapamycin (mTOR) complex 2. This results in stabilisation of T308 phosphorylation and full activation of the protein [15,16,17].

AKT localisation and activation is terminated through removal of the 3′ phosphate from PIP3 by the lipid phosphatase PTEN (phosphatase and tensin homolog) making it an important tumour suppressor [18]. Mutations causing the loss or inactivation of PTEN have been shown to lead to the development of cancer due to the hyperactivation of the PI3K pathway [19].

Over one-hundred proteins have been shown to be directly phosphorylated by AKT with many of these regulating key downstream signalling nodes (excellently reviewed in Manning et al. [20]). For the purposes of this review, we will focus specifically on those substrates and networks that have been recognised as essential for PI3K-mediated feedback loops and mechanisms of resistance to PI3K inhibitors, although it is very likely that more will be identified in the near future. Specifically, AKT phosphorylates proteins involved in cell cycle and glycogen synthesis (GSK3), glucose uptake (AS160), cell survival (FOXO, BIM, BAX, BAD and BCL2), and protein synthesis and proliferation (TSC2 and PRAS40) (Figure 1). The most well studied of these signalling cascades comprises members involved in mTORC1 activation. This involves a relay whereby AKT phosphorylates and inhibits the tuberous sclerosis complex 2 (TSC2). The C-terminal domain of TSC2 functions as a GAP for the RAS-related GTPase (RHEB) promoting the conversion of active RHEB-GTP to inactive RHEB-GDP [21]. The phosphorylation of TSC2 by AKT results in the rapid release of the TSC complex from RHEB permitting RHEB to become GTP-loaded activating mTORC1. AKT simultaneously phosphorylates the mTORC1 inhibitory subunit proline-rich AKT substrate of 40 kDa (PRAS40) at threonine 246 (T246) to a level of regularity that PRAS40 T246 is considered a proxy for AKT activation in most cells and tissues [22]. Although the biological function of this phosphorylation remains unclear PRAS40 phosphorylation results in the dissociation of PRAS40 from mTORC1, which increases mTORC1 binding to other substrates [23].

mTOR is a serine–threonine kinase that forms a part of the phosphatidylinositol kinase-related kinase (PIKK) family of kinases. mTORC1 is comprised of three core components: mTOR catalytic subunit, regulatory associated protein of mTOR (Raptor) and mammalian lethal with Sec13 protein 8 (mLST8/GβL) [24,25,26]. In addition to these core components the mTORC1 complex also contains PRAS40 and DEP domain containing mTOR interacting protein (DEPTOR), which act to inhibit the complex [22,27]. Similar to mTORC1, mTORC2 contains two of the three core proteins but instead of RAPTOR, mTORC2 forms a complex with rapamycin insensitive companion of mTOR (RICTOR) [28]. Likewise, mTORC2 contains DEPTOR as well as the regulatory subunit SIN1 [29]. 

mTORC1 regulates several cellular processes which are implicated in homeostasis and cell growth, which include autophagy, lipogenesis, cellular proliferation, cell survival, glucose metabolism and protein synthesis [21,30]. The two most characterised targets of mTORC1 are eukaryotic initiation factor 4E-binding protein 1 (4E-BP1) and ribosomal protein S6 kinase 1 protein (S6K1), which are both involved in protein synthesis [31]. 4E-BP1 is hyperphosphorylated by mTORC1, inhibiting its binding to eukaryotic initiation factor 4E (eIF4E), leading to cap-dependent translation of key cell cycle regulators such as MYC or CYCLIN D1 [32]. 4E-BP1 is known to inhibit mRNA translation in healthy cells; however, it has been seen to be overexpressed in numerous human cancers and has been linked with a poor prognosis in numerous cases [33,34,35].

S6K1 (p70S6Kα) or p70S6 kinase is also phosphorylated by mTORC1, which results in the downstream phosphorylation of S6 ribosomal protein (rpS6) [36]. S6K1 is a serine/threonine kinase that can regulate cellular mechanisms heavily involved in oncogenesis such as cellular proliferation, growth, migration, invasion, survival, apoptosis and protein translation [37,38,39]. S6K1 does this by phosphorylating proteins that are involved in translation and protein biosynthesis [40]. S6K1 is comprised of five domains: an autoinhibitory domain, linker domain, N-terminal regulatory domain, catalytic domain and the C-terminal domain [41].

The activity of S6K1 is sequentially regulated by the phosphorylation of various serine/threonine sites within its C-terminal domain by p38, ERK and CDC2, resulting in the release of the catalytic domain from C-terminal inhibition [42]. The hydrophobic motif phosphorylation site on S6K1 is then phosphorylated by mTORC1 at T389, followed by phosphorylation of the T229 site in the activation loop by PDK1 [43]. This results in the full activation of the protein permitting downstream phosphorylation of rpS6 and protein synthesis.

Another target of the mTORC1-S6K1 cascade is insulin receptor substrate 1 and 2 (IRS1 and IRS2). As part of a negative feedback loop, phosphorylation of these proteins by mTORC1 and S6K at multiple serine residues results in the rapid degradation of both proteins stifling upstream PI3K signalling [44]. Taken together, AKT phosphorylation results in activation of a number of downstream signalling networks, which, along with other key signalling pathways, form the basis for cellular proliferation and growth. This is due to these components being constitutively activated, as a result of mutations that drive tumorigenesis.

## 3. Pharmacological Targeting of the PI3K Pathway

From early on in its discovery it became readily apparent that the PI3K pathway is important for translating external environmental cues to cellular actions required for facilitating growth and cell cycle progression. The fact that activating alterations in the PI3K pathway were frequently found in a variety of cancers further cemented the notion that cancer cells needed to exploit the PI3K downstream pathways to initiate and maintain tumorigenesis. Importantly, these findings suggested that this class of enzymes are a prime target for chemotherapy agents.

The first PI3K inhibitor, wortmannin, was derived from fungi metabolites in 1987 and initially was found to negatively affect oxidation in neutrophils [45]. Further investigation identified that wortmannin non-specifically reacted with various elements in the PI3K pathway, including PI3K and mTOR as well as an array of other proteins [46,47]. However, the compound effects on the pathway were irreversible, toxic and unstable when used in animal studies, thus limiting its use as a targeted therapy [48,49]. LY294002 was the first artificial inhibitor of PI3K, developed to combat the harsh effects of wortmannin while still maintaining the inhibitory effects on multiple proteins in the PI3K pathway including the PI3K complex, mTOR and DNA-PK [50]. LY294002 is a slightly more stable compound and its effects on PI3K are reversible. However, the future of this drug was limited to pathway analysis over clinical development due to constraints with its pharmacological properties [48,51]. Other inhibitors that targeted multiple PI3K proteins were developed that share similar structures to wortmannin and LY294002, including SF1126 and PX-866 [49,52]. Although both analogues presented better pharmacologic properties than their predecessors, only PX-866 entered early phase clinical trials, with mixed results.

A second seminal discovery was the isolation of rapamycin, a macrolide produced by *Streptomyces hygroscopicus* bacteria taken from a soil sample at Easter Island (also known as Rapa Nui). Rapamycin (Sirolimus) was demonstrated to have broad antiproliferative activities potentially through inhibition of the S6K1, a downstream mediator of PI3K signalling. However, it was not until the early 1990s that mTOR was identified as the target of the toxic effects of rapamycin [53,54]. Although rapamycin-based therapies have demonstrated some benefits in clinical settings, the use of rapamycin as a single agent is limited due to its modest efficacy, this is primarily due to its inability to completely abrogate mTORC1-mediated signalling events. However, as a scientific tool, the use of rapamycin has led to some of the greatest contributions to our understanding of the intricacies involved in targeting the PI3K-AKT-mTOR pathway.

A decade ago, there were more then 30–40 compounds in preclinical development targeting various nodes in the PI3K pathway with all but a few of these receiving FDA approval in recent years. These inhibitors were classed into six general classes: rapamycin analogues, active site mTOR inhibitors, pan-class I PI3K inhibitors, isoform-selective PI3K inhibitors, dual inhibitors and AKT inhibitors [55]. These different classes are discussed briefly below. For a complete overview of PI3K pathway inhibitors in clinical trials we refer to Janku et al. [56,57].

### 3.1. mTOR Inhibitors

Following on from the early success of rapamycin in certain clinical settings an enormous effort was made to effectively target mTOR in cancer. Unsurprisingly, rapamycin has been the muse in the development for an array of anticancer drugs with improved pharmacokinetic properties (Table 1). Note that rapamycin is not a direct inhibitor of mTORC1 but rather rapamycin binds to FK506-binding protein of 12kDA (FKBP12) which then interacts with the FKBP12-rapamycin binding domain (FRB) of mTOR, thus inhibiting mTORC1 functions [58]. In contrast, the FKBP12-rapamycin complex cannot physically bind to mTORC2, limiting the acute inhibition of mTORC2. As such, mTORC1 and mTORC2 are designated respectively as rapamycin-sensitive and rapamycin-insensitive complexes. There are currently two types of inhibitors in this category: allosteric mTORC1 inhibitors or the ATP-competitive catalytic mTOR inhibitors [59].

#### 3.1.1. Allosteric mTOR Inhibitors

Temsirolimus (cell cycle inhibitor-779, CCI-779) is a derivative of rapamycin and is currently approved for the treatment of advanced-stage renal cell carcinoma. Although temsirolimus monotherapy prolonged overall survival, objective response rates are low. Interestingly, a retrospective study identified a number of risk criteria that were directly associated with poor prognosis [56].

Another derivative of rapamycin is everolimus (RAD001). Unlike temsirolimus, everolimus is administered orally [60]. Despite having low response rates as monotherapy towards some tumours, everolimus has been FDA approved for the treatment of advanced renal cell carcinoma, hormone receptor-positive/HER2-negative (HR+/HER2-) breast cancer, as well as an array of neuroendocrine tumour types [61,62,63]. Importantly, findings from the BOLERO-2 trial demonstrated that treatment with everolimus in conjunction with aromatase inhibitor exemestane significantly improved the overall response rate in patients with HR+ advanced cancer that had previously been treated with nonsteroidal aromatase inhibitors, and its activity was irrespective of PI3K genetic alterations [64]. These clinical findings validated the notion that the PI3K pathway is a major contributor to the development of resistance to hormone therapy.

Finally, ABI-009 (*nab*-sirolimus) has recently been submitted for approval for the treatment of perivascular epithelioid cell neoplasms (PEComa). Nab-sirolimus is an injectable nanoparticle form of human albumin-bound sirolimus which has demonstrated increased tumour uptake. Preclinical studies have demonstrated that nab-sirolimus indicated increased tissue penetration and pharmacodynamics compared to equal doses of oral mTOR inhibitors [56].

#### 3.1.2. Active Site or Catalytic mTOR Inhibitors

Catalytic mTOR inhibitors target the kinase domain of both mTORC1 and mTORC2 (and subsequently, AKT), making them more efficient at inducing an inhibitory effect over the current allosteric therapies [65]. The most promising ATP-competitive mTOR inhibitors in development include MLN0128 (sapanisertib), PP242 (tokinib), AZD2014 (vistusertib) and AZD8055. These therapies have shown promising results in a range of cancer types, and have been shown to reverse incidences of resistance to other therapeutics [66].

In clinical trials, MTOR inhibitors are generally well tolerated with the most common side effects being headache, fatigue and erythema (skin rash). However, the use of MTOR inhibitors is known to be associated with a higher risk of developing Hypertriglyceridemia, hypercholesterolemia and hyperglycaemia [67,68,69,70].

### 3.2. Dual mTOR and PI3K Inhibitors

During the early developmental phases of mTOR and PI3K inhibitors it was noted that these enzymes possessed structural similarities in their kinase domains making it relatively easy to design ATP-competitive drugs targeting both kinases simultaneously [71,72]. Conceptually, the main benefit of dual-targeting these particular elements is due to a number of reports which demonstrated that mTOR inhibition results in repression of a negative feedback loop which activates the PI3K and MAPK pathways. As such, inhibition of both PI3K and mTOR was thought to limit these compensatory mechanisms [73,74]. The first of these dual inhibitors BEZ235 (dactolisib) was created by Novartis Pharmaceuticals initially as a pan-PI3K inhibitor, but studies also demonstrated increased binding efficacy for mTOR [75,76,77]. The drug showed promising anticancer results in preclinical studies, resulting in its progression to early phase clinical trials. Thus, making it the first of many PI3K pathway inhibitors to do so [78]. Despite the overwhelming hype over this treatment option, BEZ235 exhibited highly toxic characteristics in patients and limited pharmacokinetic properties, and thus was not deemed a worthwhile treatment option on its own [79]. Subsequently, other PI3K/mTOR dual-targeting inhibitors include GDC-0980 (apitolisib), XL765, PF-04691502 (gedatolisib), PF-05212384 (PKI-587) and GSK2126458 (omipalisib), and they all showed favourable results in preclinical studies but alas faltered in clinical trials in terms of both toxicities and percentage effectiveness in patients (Table 1) [48,57]. A bright spot for this family of inhibitors may be GDC-0084 (paxalisib) which has been shown to delay the progression of newly diagnosed glioblastoma, with manageable adverse side effects [80,81]. Currently, there are no dual inhibitors approved for clinical use.

### 3.3. Pan-Class I PI3K Inhibitors

Further insights into the molecular characteristics of the PI3K pathway elements led to the development of inhibitors that can specifically target individual molecules and their associated isoforms. The premise behind the development of these agents was that they would be effective in cancers with elevated levels of PIP3, regardless of the tumours bearing aberrant PI3K mutations or PTEN loss. Some of these pan-PI3K inhibitors include BKM120 (buparlisib), GDC-0941 (pictilisib) and BAY 060-6946 (copanlisib), which are three of the most well studied inhibitors; both in a preclinical setting and in current clinical trials [82,83]. Buparlisib showed promising results in patients with both triple-negative breast cancer (TNBC) and ER-positive breast cancers, especially when used alongside other inhibitors such as the PARP inhibitor olaparib or the oestrogen receptor antagonist, fulvestrant [84,85,86]. Buparlisib has the ability to permeate the blood–brain barrier (BBB) suggesting it could be beneficial for targeting various brain malignancies, but has the tendency to cause mood changes [83,87]. Pictilisib, on the other hand, barely penetrated tumours in models with intact BBB potentially explaining the lack of associated mood changes [80,88]. Both buparlisib and pictilisib have had less success in terms of clinical trials; however, both display favourable pharmacokinetic properties suggesting combination therapies may be a potential future use of this compound [82,89,90].

Interestingly, due to its targeting of PI3K alpha/delta copanlisib has shown good anti-tumour properties in patients with non-Hodgkin lymphoma, and was FDA approved for patients with refractory follicular lymphoma in 2017 [91]. Some positive responses were also observed when copanlisib was used simultaneously with cisplatin plus gemcitabine (CisGem) in patients with advanced cancers despite more recent phase II studies which did not met their primary endpoint [92].

Duvelisib (IPI-145) is an inhibitor that targets the PI3K isoforms delta and gamma, and exhibits anticancer activity in primary chronic lymphocytic leukaemia (CLL) cells while normal B cells remain unscathed by its cytotoxic effects [93]. Various clinical trials have shown duvelisib to have a satisfactory safety profile with tolerable adverse effects in patients with varying types of lymphoma as well as demonstrating relatively high response rates [94,95,96]. The FDA approved duvelisib for use in patients with refractory CLL or small lymphocytic lymphoma following the use of a minimum of two other systemic treatments and was fast-tracked for approval for patients with refractory follicular lymphoma [97]. As it stands copanlisib and duvelisib are the only two pan-PI3k inhibitors which have been approved for clinical use.

Other pan-PI3K inhibitors which were evaluated in clinical trials include GDC-0032 (taselisib) and XL-147 (pilaralisib). Taselisib can target all isoforms of p110 subunit of PI3K with high specificity, with the exception of the beta isoform [57]. It has shown a promising response rate in patients; however, it does seem to cause borderline adverse effects, including stomatitis and diarrhoea [98]. In the recent SANDPIPER trial, the results indicated that taselisib in combination with fulvestrant for patients with PIK3CA mutated, HR+/HER2-negative breast cancer, met its primary endpoint of increased progression-free survival. Unfortunately, it was concluded that given its safety profile and modest clinical activity this combination has no clinical utility [99]. Pilaralisib targets all isoforms of PI3K and has been found to inhibit PIP3 formation, thus preventing the phosphorylation of AKT and downstream S6 in preclinical models of various cancers [100]. Pilaralisib was assessed in phase II clinical trials for endometrial cancers by monotherapy, as well as in patients with breast cancers in conjunction with HER2 antagonist trastuzumab or aromatase inhibitor, letrozole [101]. Although in both cases pilaralisib, either as monotherapy or in combination, was well tolerated, overall response rates were low and the drug has been discontinued.

### 3.4. Isoform-Selective PI3K Inhibitors

Like dual PI3K-mTOR inhibitors clinical development of pan-Class I PI3K inhibitors has been greatly limited by the dose-limiting toxicities observed at the required doses needed for adequate target engagement. This, above all else, has led to rapid development of isoform-selective PI3K inhibitors which are predicted to have a wider therapeutic index as their genetic alterations are limited to primarily malignant cells (Table 1). The most prominent illustrative example is CAL-101 (idelalisib), a potent inhibitor of the delta isoform of PI3K. Unlike isoforms p110α and p110β which are ubiquitously expressed in all tissues, p110δ is primarily expressed in leukocytes, limiting any deleterious effects on PI3K signalling in healthy tissues [102,103]. p110δ is in integral component of B cell receptor signalling in chronic lymphocytic leukaemia (CLL) and downregulation of p110δ with idelalisib in combination with rituximab demonstrated significant overall responses rates with progression free and overall survival benefit [104]. This led to idelalisib being the first isoform-selective PI3K inhibitor to be FDA approved. Current clinical trials are assessing the synergistic use of CAL-101 alongside other inhibitors in an array of cancers and other diseases.

The most recent isoform specific therapy sparking interest is the p110α-specific inhibitor, BYL719 (alpelisib) [105]. In initial phase I studies, alpelisib monotherapy demonstrated promising clinical activity; however, subsequent data revealed that PI3K inhibition enhances oestrogen signalling in HR+ breast cancer, limiting the overall effectiveness of these compounds [106,107,108]. Alpelisib has been approved in combination with fulvestrant for the treatment of metastatic breast cancer in patients harbouring alterations in *PIK3CA*. Thus, making alpelisib the first isoform-specific drug targeting p110α to be FDA approved in patients under the aforementioned conditions [109,110].

### 3.5. AKT Inhibitor

An alternative target for downregulation of the PI3K-AKT-mTOR pathway is AKT itself. AKT inhibitors are separated into two differing groups; the allosteric inhibitors which target the PH domain of AKT and the ATP-competitive inhibitors which are selective for the catalytic site of AKT (Figure 2) [111,112,113]. Specific catalytic inhibition of AKT was predicted to be difficult as the ATP binding pocket of AKT shows a high degree of homology with protein kinase A and protein kinase C. However, these reservations have been overcome using allosteric inhibitors of AKT. The pan inhibitor MK2206 remains the most prominent of the allosteric inhibitors, however others such as TAS-117 have also shown promising effects [56,114]. The effectiveness of this targeted therapy is attributed to its targeting of the PH domain on AKT, which prevents its activation and translocation to the cell membrane [115]. MK2206 has displayed prominent antitumour activity in preclinical breast cancer models with *PIK3CA* and/or *PTEN* mutations both on its own, or when enhanced with endocrine therapies [116]. However, MK2206 has not progressed further in these clinical scenarios. At present MK2206 is currently only being assessed in a phase II clinical trial in combination with the EGFR inhibitor gefitinib for patients with non-small cell lung cancer (ClinicalTrials.gov Identifier: NCT01147211).

TAS-117 is a selective oral allosteric AKT inhibitor, which has recently been established as a potent inhibitor of cellular growth in multiple myeloma [117]. TAS-117 is currently being evaluated in patients with solid tumours harbouring mutations in the canonical PI3K pathway.

Similarly, preclinical work utilising ATP-competitive therapies demonstrated promising activity in a variety of tumour types including lung, breast, prostate and melanoma cells. Most notably of these is GDC-0068 (ipatasertib). Ipatasertib in combination with endocrine therapy and CDK4/6 inhibitors is currently in registrational studies for the treatment of metastatic breast cancer (ClinicalTrials.gov Identifier: NCT03959891). In earlier studies ipatasertib was well tolerated and importantly clinical activity was demonstrated in a significant proportion of patients treated [118]. Importantly, ipatasertib has demonstrated durable clinical responses in a patient with the AKT hyperactivating mutation E17K [119]. Ipatasertib is also being evaluate in combination with abiraterone acetate in patients with castration-resistant prostate cancer (NCT01485861) and in combination with the PARP inhibitor Rucaparib in patients with advanced breast, ovarian or prostate cancer (ClinicalTrials.gov Identifier: NCT03840200). Likewise, AZD5363 has depicted promising results. In a variety of patient tumours containing the AKT E17K mutation, AZD5363 treatment resulted in cessation of tumour growth, and in some cases, tumour shrinkage [120]. Like ipatasertib, AZD5363 is currently being evaluated in a late-stage clinical trial for patients with triple-negative breast cancer (ClinicalTrials.gov Identifier: NCT03997123).

After two decades of research into the clinical translatablility of PI3K pathway inhibitors, it is reasonable to state that despite modest clinical activity in solid tumours PI3K inhibitors have the potential to be important pharmacological compounds for the treatment of these tumours. However, at this stage there is a requirement for an increased understanding in PI3K biology, as well as the intrinsic processes which are deregulated following PI3K inhibition, to limit the overall sensitivity of these compounds. A few of these intrinsic processes are highlighted below.

## 4. Reactivation of PI3K Signalling

The inability of compounds targeting the PI3K pathway to combat cancer growth and metastasis effectively, has only emphasised the fact of how extensive PI3K/AKT/mTOR signalling can be. In general, studying the effects of various targeted therapies has shown how cancers can disregard the downregulation of the pathway by either maintaining or re-establishing activation of the targeted pathway or by inducing alternate signalling pathways. This is occasionally determined by genetic predisposition, whereby the selection of rare pre-existing resistance-conferring genetic mutations limit overall therapy sensitivity. In some of these cases, the development of drug resistance may be mediated by loss of PTEN expression; altered expression of RSK3/4, PIM, AXL, FOXM1, NOTCH, c-MYC, PDK-1-SGK1, SGK3 and CDK4/6; or KRAS mutations [77,121,122,123,124,125,126,127,128,129,130]. However, under certain contexts downregulation of the targeted pathways can result in genetically independent intrinsic compensatory mechanisms with survival rather than continued proliferation being the ultimate outcome. These compensatory mechanisms can also be referred to as feedback loops or adaptive responses.

To accurately translate cues from the extracellular environment into appropriate transcriptional outputs, signal transduction is tightly regulated by a number of compensatory mechanisms that can either effectively downregulate or upregulate signal strength [131]. The mechanisms to maintain this tight balancing act are retained in cancer cells, but in the majority of cases their importance is diminished in the presence of oncogenic driver mutations. Nevertheless, these feedback loops are once again activated or derepressed upon targeted inhibition leading to reactivation of the pathway and eventual therapy evasion.

A number of recent studies have not only shown that short-term treatments can result in the activation of these feedback loops subsequently decreasing overall response rates, but that chronic administration of these therapies may result in the establishment of a reservoir of slow cycling cells that eventually may acquire resistance-conferring genetic mutations [132]. The most commonly identified mechanisms of resistance to targeted therapies are discussed below.

### 4.1. Insulin Signalling and PI3K Reactivation

To continuously meet the needs required for hyperproliferation, transformed cells must appropriately adjust their signalling and metabolism. The PI3K pathway is one of the key pathways that regulates nutrient uptake and cell survival. It is also therefore no surprise that the PI3K signalling cascade is one of the most frequently mutated pathways in cancer.

Altered glucose metabolism is a consistent feature in cancer cells, characterised by an increased rate of glucose uptake and a glycolytic conversion to lactate. This phenomenon was originally observed by Otto Warburg over one-hundred years ago and is referred to as aerobic glycolysis or the “Warburg effect”. In addition to providing numerous metabolic intermediates required by various metabolic pathways and cellular processes, such as protein production, it also supplies the cancer cell with its energy requirements through the generation of ATP [133].

The uptake of glucose into cells is primarily mediated by the glucose transporter family (GLUTs) [134]. Constitutive activation of downstream AKT signalling has been shown to be sufficient to induce aerobic glycolysis by promoting glucose uptake through GLUT1 and GLUT4. Notable mechanisms promoting this involves AKT2-mediated phosphorylation of the Rab-GTPase-activating protein TBC1D4, also known as AS160. Specifically, AS160 impairs GLUT4 migration to the cell membrane. AKT2 phosphorylation blocks AS160 function enhancing GLUT4 intracellular vesicular transport, and as such promotes glucose uptake following insulin release [135]. However, this mechanism appears to be specific for GLUT4, and is thus unlikely to play a major role in glucose uptake into cancer cells, which predominantly express GLUT1. The transport of GLUT1 to the cell membrane has been associated with AKT activation, although the mechanism of this is not fully understood [136,137,138,139]. Nevertheless, recent reports have demonstrated that thioredoxin-interacting protein (TXNIP) is a direct target of AKT, and the phosphorylation of which annuls TXNIP-mediated endocytosis of GLUT1 and GLUT4, resulting in a pronounced increase in glucose uptake. The transcription of GLUT1 occurs through downstream effectors of the PI3K pathway, including through c-Myc, mTORC1-induced HIF1α expression, and various transcription factors associated with glucose metabolism [139,140,141,142].

Apart from glucose transporter guidance it has been shown that PI3K-AKT pathway regulates multiple nodes of the glycolytic cascade (reviewed by Hoxhaj et al. [133]). As indicated above, the PI3K pathway is stimulated by an array of growth factors that target cell surface receptors in order to achieve, among other things, continual metabolic requirements of the cell. This includes activation of the epidermal growth factor receptor (EGFR), platelet derived growth factor receptor (PDGFR), insulin receptor (INSR) and the insulin-like growth factor receptor (IGFR), with each of these receptors regulating pathways involved in proliferation, migration, metabolism and cell survival. As such, the interactions between these receptors and their signalling cascades can overlap to conform to synergistic responses, where, for instance, the induction of cell cycle progression also promotes an influx in energy synthesis by altering metabolic activity [143].

Mechanistically, the activation of receptors INSR and IGFR following growth factor stimulation results in the recruitment and subsequent phosphorylation of insulin receptor substrate (IRS) adaptor molecules, mainly IRS-1. The phosphorylated variants of IRS-1 can then activate PI3K through binding to both SH2 domains on the p85 subunit of PI3K, relieving its hold on the catalytic p110 subunit, promoting activation of the pathway [57,69,144,145,146]. Conversely, p85 on its own has also been found to form a sequestration complex with IRS-1 when in excess to the p110 subunit. This complex causes the relocation of phosphorylated IRS-1 into the cytosol, away from the PIP3 production site at the cell membrane. This essentially prevents IRS-1 from carrying out its intended function of pathway activation [147]. This mechanism can not only lead to the suppression of insulin signalling through the maintenance of p85 basal inhibition on p110, but can also result in insulin resistance if expressed in surplus [147,148]. Once activated, the p110α subunit of PI3K can then mediate the intracellular response to insulin stimulation, promoting growth and glucose homeostasis in most tissues [149]. As such, aberrations to the *PIK3CA* gene encoding the p110α subunit are associated with various diseases and pathologies associated with glucose metabolism and tissue growth [143].

Two of the most common adverse effects witnessed in patients undergoing PI3K inhibitor clinical trials are hyperglycaemia and hyperinsulinemia, which are readily managed in patients under circumstances where they remain chronic [57,69,144,145,146]. Upon treatment with PI3K/AKT inhibitors, glycogenolysis in the liver is promoted and the ability of AKT2 to regulate GLUT transport in adipose tissue is downregulated. This causes the levels of glucose taken up in these cells to be reduced, prompting the transient development of systemic hyperglycaemia. However, this effect tends to be transient. As part of the body’s normal glycaemic regulation, insulin production is increased in the β cells of the pancreas following high blood glucose levels. This response returns blood glucose to within the normal range; however, this can also result in systemic hyperinsulinemia [150]. Because of its function within the PI3K pathway, increased insulin is likely to stimulate insulin receptor (IR)-rich tumours promptly activating downstream signalling and tumour maintenance (Figure 2).

This effect can be seen when looking at glucose uptake using ^18^F-deoxyglucose positron emission tomography (FDG-PET) in tumours following PI3K inhibitor treatment. In a phase Ib trial combining buparlisib with letrozole, half of the treated patients exhibiting a marked reduction in FDG tumour uptake correlating with clinical benefit, whereas those with increased FDG uptake correlated with tumour progression [110]. These data indicate that an observed increase in tumour FDG uptake shortly after PI3K inhibitor treatment is potentially explained by an insulin surge following PI3K inhibition. This indicates that patients are less likely to respond to the treatment offered. This effect is possibly due to increased IR expression within the tumours, suggesting these results as likely biomarkers associated with patient responses following treatment [151]. It has also recently been reported that lactate can serve as surrogate to measure glycolytic flux and can effectively determine PI3K pathway inhibition [125,152].

Importantly, Hopkins et al. studied the effects of negating the insulin response following PI3K/AKT inhibitor treatment in animal models through both pharmacological and dietary influences [149]. The authors reasoned that using the antidiabetic drug metformin and compounds targeting the sodium glucose cotransporter 2 (SGLT2), would lower insulin levels/increase insulin sensitivity and reduce glucose reabsorption at the kidneys, respectively. The efficacy of a ketogenic diet was also assessed, which utilises glycogen stores, preventing a surge of glucose release from the liver and reducing overall blood glucose levels and increasing insulin sensitivity. Interestingly, both SGLT2 inhibition and the ketogenic diet showed promising results reducing glucose levels and lowering insulin release upon treatment with buparlisib. Promising data were also reported with the PI3K pathway inhibitors alpelisib, pictilisib, taselisib, GDC-0098 and Copanlisib in murine models placed under a ketogenic diet [149]. Translation of these methods in clinical trials will further assess the use of SGLT2 and ketogenic diets in improving the sensitivity of inhibitors in patients and will further determine the effects of increased efficacy of these drugs in the clinical setting.

As insulin is a potent instigator of PI3K pathway activation, it is not surprising that not only maintenance of insulin signalling but also reactivation of the components of this pathway through feedback loops have been described as critical mediators of response to PI3K pathway inhibitors. The first observations that downstream signalling could regulate PI3K activity through a feedback loop came from a chain of studies that revealed that insulin treatment led to the phosphorylation and subsequent proteasomal mediated degradation of the adapter protein IRS-1 [153,154,155]. Subsequently, a number of studies demonstrated that this effect was mediated through both transcriptional and post translational mechanisms. This is not altogether surprising as IRS-1 is a critical regulator of PI3K signalling and therefore to maintain the desired levels of PI3K pathway activation diminished levels of IRS-1 levels would ensure that prolonged hyperactivation of PI3K signalling does not occur. Unfortunately, the reverse also holds true. In a seminal paper by Carracedo et al. [44], it was shown that pharmacological inhibition of the PI3K/AKT/mTOR pathway with the mTOR inhibitors rapamycin and everolimus released this negative feedback loop resulting in reactivation of the PI3K and MAPK pathways (Figure 2) [44]. This mechanism, although controversial at the time, sparked the inquest into the use of synergist inhibitors which target multiple elements of the pathway in an attempt to limit this regulatory activity. The relevance of these findings was further underscored by work in a number of labs which demonstrated that PI3K/AKT/mTOR pathway inhibition leads to the upregulation of a number of RTKs (discussed in detail below).

### 4.2. Receptor Tyrosine Kinase Reactivation

IGF-1R, EGFR, HER2 and HER3 are just some RTKs that are stimulated following inhibitor treatments promoting the activation of the PI3K and MAPK signalling cascade [77,156,157]. This event was found to be caused by the loss of the inhibitory effect AKT has on the Forkhead Box O (FOXO) family of transcription factors and mTORC1, promoting the transcription of these RTKs and reactivation of the signalling cascade [74,156,157,158]. When the PI3K pathway is stimulated, AKT phosphorylates FOXO proteins within its nuclear localisation sequence creating a 14-3-3 binding site masking the nuclear localisation signal and preventing nuclear translocation. The sequestration of FOXO in the cytoplasm attenuates the transcriptional capabilities of the FOXO proteins [159]. Through this process PI3K hyperactivation suppresses the induction of a number of FOXO targets involved in the induction of apoptosis or cell cycle arrest. Likewise, it inhibits the ability of FOXO to upregulate the transcription of a number of RTKs. Therefore, this mechanism acts as an indirect feedback mechanism limiting the overall activation of the RTKs by extracellular stimuli. However, upon PI3K inhibition either through loss of stimulus or through treatment with chemical compounds targeting this pathway, FOXO phosphorylation is blocked. This enhances FOXO expression in the nucleus resulting in the activation of a number of RTKs and partially restoring PIP3 activity (Figure 3). Therefore, PI3K pathway activation can never be fully attenuated as PIP3 levels will be maintained, driving cell survival. This was further elegantly demonstrated by Muranen and colleagues using ovarian cancer 3D spheroids treated with dactolisib where they demonstrated that PI3K pathway inhibition enhances apoptosis exclusively in the inner matrix-derived cells, while outer matrix-derived cells continued to display low levels of proliferation as marked by Ki67 staining [160]. Using reverse phase protein array analysis (RPPA), they demonstrated that BEZ235 treatment decreased phosphorylation of FOXO while concomitantly enhancing the expression and activation of various RTKs including EGFR, HER2, c-KIT and IGF1R [160]. Furthermore, in a separate study, Lin and colleagues demonstrated that FOXO can upregulate the mTORC2 component RICTOR resulting in increased AKT S473 phosphorylation [161].

The mechanism of AKT reactivation in these above-mentioned models is mediated by PI3K p110α. However, in PTEN-null tumours PI3K p110β is the primary PI3K isoform driving PI3K signalling [162]. This discrepancy in PI3K activation through various cellular contexts highlights the many altered signalling routes that can lead PI3K mediated tumour proliferation. Moreover, it reasons that because these mutations affect various nodes of the same signalling cascade isoform-specific PI3K inhibitors are unlikely to function in all genetic contexts. This was demonstrated in two seminal papers by Costa et al. and Schwartz et al. [163,164]. Costa et al. demonstrated that inhibition of HER2-amplified cell lines with alpelisib resulted in a rebound of PIP3 levels after 6 h [163]. This rebound was dependent on p110β. Interestingly, in PIK3CA-mutant tumours, this rebound was not observed. Likewise, Schwartz et al. [164] demonstrated that treatment of PTEN null cells with the p110β inhibitor AZD8186 significantly decreased PI3K signalling and tumour cell growth. In these models, selective p110α inhibition had no effect, as these tumours are solely dependent upon p110β signalling. However, AKT/mTOR downregulation in these models was transient because downregulation of mTOR led to derepression of FOXO and subsequent RTK transcription leading to p110α mediated AKT signalling. In both studies, targeted inhibition with isoform specific PI3K inhibitors (p110α or p110β) triggered AKT/mTOR signalling, through the reactivation of the other PI3K isoform. Importantly, this reciprocal activation was annulled through the concomitant inhibition of p110α and p110β, resulting in greater antitumour activity compared to either inhibitor alone.

More recently, it has been demonstrated that as part of a negative feedback loop mTOR regulates PTEN translation through 4E-BP1 [165]. The overall effect of this is to limit excessive PI3K signalling through the targeted regulation of PIP3 levels in the cells. Conversely, downregulation of PI3K signalling blocks mTOR phosphorylation of 4E-BP1 inhibiting PTEN translation, resulting in a rebound of AKT phosphorylation as soon as 2–4 h post-treatment. A number of interesting conclusions were derived from these studies. The first was that PTEN downregulation was predominantly observed following PI3K inhibition with p110α, but not p110β inhibitors, even though treatment with the p110β inhibitor AZD8186 effectively downregulated AKT signalling and a similar rebound in AKT signalling was observed at earlier time points. Critically, no downregulation of mTOR targets was observed upon treatment with AZD8186 at these early time points, which likely explains the lack of PTEN downregulation. Nevertheless, it will be interesting to see if the AZD8186-mediated AKT rebound is dependent upon FOXO downregulation as determined by Schwartz et al. [164] or if alternate rebound mechanisms exist. Importantly, these results also give credence to earlier observations by Costa et al. [163] and may suggest the mechanism of action for the increase in PIP3 levels following alpelisib treatment in this previous work. Furthermore, this work confirms the observations in other studies, which advocate that variations in oncogenic mutations in the PI3K pathway significantly alters the levels of mTOR activity. It has been previously noted that cell lines harbouring PIK3CA mutations display lower levels of AKT phosphorylation compared to cell lines with PTEN loss [166]. In addition, cell lines with PTEN loss and activating mutations in PIK3CA do not confer the same level of AKT/mTOR signalling as cells with PTEN loss and coexistent PI3K activation mediated by upstream regulators (ex. RTKs) [165]. This is further confounded by the fact that cell lines with oncogenic helical mutations in PIK3CA display no demonstrable activation of AKT, even though they display equivalent levels of PDK1 expression and activation [166]. To circumvent this deficiency, PIK3CA mutant cell lines activate downstream mTOR signalling through the serum and glucocorticoid regulated kinase 3 (SGK3). SGK3 shares a 50% identity with the catalytic domain of AKT and can phosphorylate a number of AKT substrates including TSC2, promoting the activation of mTORC1. Downregulation of SGK3 by both genetic and chemical means significantly diminishes the viability of PIK3CA mutant cancer cells highlighting the importance of this signalling node in tumours with low AKT activation [166,167]. Although SGK3-mediated signalling is not rapidly activated through a compensatory mechanism following PI3K pathway inhibition, prolonged AKT inhibition does increase SGK3 mRNA levels over a number days [167]. The precise mechanism of SGK3 induction remains unknown; however, SGK3 has been reported to be an oestrogen transcriptional target in ER+ breast cancer suggesting that low levels changes in oestrogen signalling may be sufficient to drive downstream mTOR activation, limiting the use of single agent PI3K pathway inhibitors (see below) [168].

Apart from transcriptional induction of respective RTKs involved in the reactivation of the PI3K-MAPK pathways reports have also indicated that PI3K/mTOR inhibition activates JAK/STAT signalling as early as 4 h post-treatment with upregulation of the pathway remaining active for up to 20 h. Interestingly, this perpetual upregulation of JAK/STAT5 appears to be regulated through biphasic and intriguingly independent mechanisms. At later time points, JAK/STAT induction is dependent upon PI3K inhibition-mediated secretion of several cytokines including IL-8 [169]. Notable secretion levels of IL-8 occurred were only detected 20 h after the addition of PI3K inhibition, suggesting that IL-8 upregulation is transcriptionally mediated. However, the authors noted that this would not explain the immediate upregulation of pJAK/STAT. This was further validated when selective inhibition of CXCR1 (the cognate receptor for IL-8) only decreased overall JAK/STAT activation at later time points. Britschgi and colleagues further demonstrate that accumulation of IRS-1 preceded JAK/STAT5 phosphorylation at earlier time points indicating that IRS-1 may be a mediator of JAK/STAT signalling resulting in an overall decrease in PI3K inhibitor sensitivity [169].

More recently, it has been demonstrated that PI3K inhibition promoted AKT reactivation by the E3 ubiquitin ligase Skp2, an effect which was diminished when either PDK-1 or mTORC2 activity was blocked [170]. Not completely unexpectantly this effect was independent of PI3K activity or PIP3 production. As SKP2-mediated ubiquitination of AKT is required for plasma membrane recruitment it is likely that SKP2 enhances the localisation of AKT with PDK1 and subsequently mTORC2 for priming and full activation of the protein [171].

Taken together these interesting results highlight the complexity of these intrinsic feedback loops to effectively downregulate PI3K signalling. Importantly, combination therapies that target PI3K, AKT or mTOR alongside various RTK molecules effectively combat this response in preclinical studies [172,173,174,175]. However, there continue to be limitations in the clinical setting due to primarily negative adverse effects experienced by the patients, as well as a lack of selectivity when it comes to available drugs that target RTKs specifically [151].

### 4.3. Endocrine-Mediated Resistance

Approximately 85% of breast cancers express the hormone receptors ER, PR or HER2, the former having been shown to strongly correlate with mutations in *PIK3CA* [176,177]. Owing to the presence of applicable compensatory mechanisms following downregulation of either estrogen or PI3K signalling it has long been suggested that that there is an important crosstalk between these oncogenic pathways. Several studies have demonstrated that enhanced PI3K signalling either through hyperactive mutations or upregulation of PI3K pathway components drives endocrine therapy resistance [178]. These initial observations lead to seminal work which demonstrated that mTOR inhibition in combination with endocrine therapy provided significant antitumour activity [64,178,179]. The ability of the ER pathway to affect the PI3K pathway following inhibitor treatment suggests there is likewise a means for PI3K pathway inhibitors to promote the activity of ER, leading to adaptive resistance to PI3K therapies. In 2015, Bosch et al. [106] identified an enhanced luminal gene expression signature controlling ER transcription in ER+, *PIK3CA*-mutation bearing breast cancer cells following treatment with therapeutic doses of PI3K or AKT inhibitors. Not surprisingly, the enhanced expression of these genes resulted in increased ER activity and decreased PI3K inhibitor sensitivity (Figure 3) [106]. Subsequent CHIP-Seq data revealed that a large proportion of ER genes contained consensus binding motifs for the transcription factors FOXA1 and PBX1 [107]. Interestingly, the proposed mechanism prompting this reaction requires activated methylated histone marks. A subsequent study by the same group demonstrated that the PI3K pathway tightly suppresses ER-mediated transcription via direct phosphorylation of the lysine methyltransferase KMT2D (also known as MLL2 or MLL4) by AKT1, sequestering its enzymatic activity and preventing the transcription of ER and related factors [107]. Upon PI3K inhibition, activation of the methyltransferase KMT2D promotes the transition of chromatin to an accessible state, which is essential for FOXA1 and PBX1 recruitment and subsequent ER activation, and PI3K inhibitor resistance (Figure 3). Interestingly, the serum and glucocorticoid-regulated kinase 1 (SGK1), a protein which shares high similarity with the catalytic domain of AKT and can phosphorylate consensus AKT motifs, functions through a negative feedback loop to downregulate KMT2D. PI3K blockade induces ER which promotes SGK1 transcription through direct binding to its promoter. This upregulation of SGK1 results in the phosphorylation and downregulation of KMT2D and loss of ER signalling [108]. Interestingly, SGK1 can also overcome AKT inhibition. Like AKT, PDK1 can phosphorylate SGK1 resulting in downstream activation of mTORC1 and decreased PI3K inhibitor sensitivity [129]. However, it is important to note that feedback loop activation between these overlapping but independent biological mechanisms is context dependent. Although both kinases share the same upstream regulators, mTORC2 and PDK1, only AKT harbours a PH domain required for plasma membrane localisation. Castel et al. indicate that in cell lines made resistant to PI3Kα inhibitors minimal SGK1 activity was detected [129]. This contrast in resistant mechanisms between PI3K inhibition and seemingly linear downstream AKT inhibition or mTOR inhibition highlights the need for a continuing increased understanding of the PI3K biology. Only when these signalling networks are understood can we design rationale therapies to be used in combination with those in neighbouring pathways to provoke more significant prolonged patient outcomes.

### 4.4. Cellular Plasticity and PI3K Inhibitor Resistance

The above-cited mechanisms of resistance are based on intrinsic changes resulting in reactivation of the PI3K pathway or subsidiary pathways involved in maintaining cellular proliferation. However, note that downregulation of a number of the RTKs involved in PI3K signalling results in the generation of genetically independent transcriptional programs resulting in “drug-tolerant” cell populations [180,181]. This state, as opposed to primary drug resistance, is one where tumour cells transiently survive but do not proliferate on treatment. Nevertheless, these drug-tolerant populations are capable of escaping initial drug therapy but importantly do not contain the genetic mechanisms to acquire full resistance required for tumour progression. Yet, they can provide a reservoir of tumorigenic slow-cycling cells from which secondary genetic mechanisms of acquired resistance can evolve [181]. Emerging evidence has demonstrated that these cells are at the core of the development of secondary resistance not only to targeted therapy but also to immunotherapy [182]. This protective measure has been shown to be associated with a phenotypic switch, commonly referred to as cell plasticity, whereby epithelial tumours progress to a more mesenchymal state [132,183,184]. This epithelial–mesenchymal transition (EMT) has long been associated with chemoresistance [185,186]. Most importantly, this drug-refractory state is reversible upon drug withdrawal emphasising the lack of genetic mutations driving this drug-tolerant state [181].

This phenotypic switch in response-targeted therapies is likely to be mediated not only by key EMT transcription factors, but also by chromatin-modifying enzymes enhancing the accessibility of key binding motifs linked to these transcription factors. The finding that methyltransferase KMT2D activity is altered following PI3K inhibition may be a potential precursor to other chromatin-modifying enzymes and the regulation of cell plasticity [107]. A further argument to this scenario is the observation that combination therapy using epigenetic inhibitors such as histone deacetylase (HDAC) inhibitors, or bromodomain and extra terminal domain (BET) inhibitors with PI3K inhibitors were effective in preclinical models [187].

Ultimately, the initial mechanisms prompting changes in EMT upon PI3K pathway inhibition remain poorly understood with PI3K signalling involved in both the activation and inhibition of a number of EMT associated transcription factors [181,188]. At this early stage, additional work will be required to tease out the specific factors downstream of RTK signalling to determine the critical nodes regulating changes in the epigenome driving EMT and PI3K inhibitor resistance.

Cancer cells placed under selective pressure can also enter a drug-tolerant state similar to an evolutionary conserved survival strategy known as diapause. Diapause is a reversible environmentally induced state that occurs when the embryo is proliferating in unfavourable conditions such as delayed blastocyte implantation [189]. This effect has been shown to be partially regulated by mTOR, whereby mTOR inhibition induces reversible pausing of blastocyte development permitting prolonged survival without appreciable cell death or cell cycle alterations [190]. Autophagy appears to be a critical mediator of diapause as autophagy is increased prior to diapause initiation. mTOR is known to inhibit autophagy by phosphorylating autophagy-related gene 13 (ATG13) and unc-51 like autophagy-activating kinase 1(ULK1) disrupting its interaction with AMPK [191].

Interestingly, Rehman et al. recently demonstrated that in patient derived colorectal cancer models treated with standard of care chemotherapy or the mTOR inhibitor INK128 cells entered a drug-tolerant state comparable to diapause [192]. This diapause-like drug-tolerant state was exemplified by mTOR downregulation and an upregulation of autophagy [192]. Upon treatment discontinuation, cell lines once again continued to proliferate, highlighting once again the plasticity of these various drug tolerant states. Furthermore, the authors demonstrated that combination treatment using chemotherapy and autophagy inhibitors significantly decreased cell survival suggesting a potential therapeutic opportunity to target cancer cells within this diapause-like state.

An important observation is that in both drug-tolerant populations, the first mediated by changes in EMT and the second mediated by induction of diapause, the cells do not enter dormancy or complete latency but rather remain in a slow-cycling state similar to that of pluripotency. This suggests that the use of differentiation therapy such as transforming growth factor β (TGFβ) or leukaemia inhibitory factor (LIF) inhibitors may be applicable in certain contexts [192]. Nonetheless, the development of these drug-tolerant populations prior to the occurrence of secondary bona fide resistance mechanisms identifies a potential therapeutic window of opportunity and a rational approach for future targeted therapies in combination with PI3K pathway inhibitors.

## 5. Conclusions

The PI3K pathway is hyperactivated in almost all cancer types with the pathway playing a key role in tumour cell proliferation and survival. The identification of selective PI3K pathway inhibitors was met with great enthusiasm. However, the results from clinical trials have been largely disappointing with the majority of these compounds not advancing to late-phase randomised trials. The overall success of these agents has been limited by a number of factors including suboptimal patient selection and subtherapeutic maximum tolerated doses limited by dose-related toxicities. This later point, highlighted in this review, is greatly influenced by intrinsic adaptive responses which re-establish pathway activation following treatment resulting in inadequate pathway inhibition and tumour progression. Furthermore, the establishment of drug-tolerance mediated by changes in cellular plasticity or diapause-like mechanisms further limits the antitumourigenic properties of these compounds. This not only underscores the argument for an increased understanding of the complexities of PI3K signalling, but also greatly supports evidence for the use of specific combinations to overcome these mechanisms of resistance. 

## Figures and Tables

**Figure 1 cancers-13-01538-f001:**
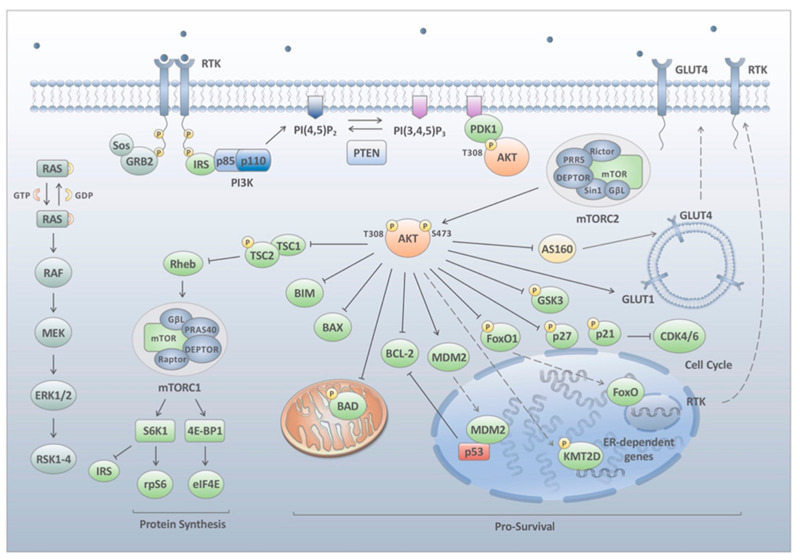
The PI3K and MAPK signalling pathways. Growth factors and other external signalling molecules stimulate the activation of the PI3K and MAPK pathways through receptor tyrosine kinase (RTK) phosphorylation at the cell membrane. Activated RTKs recruit molecules bearing phosphotyrosine-binding (PTB) or Src homology-2 (SH2) domains, such as IRS or p85, respectively. For PI3K enzymes, the binding of the p85 subunit activates the catalytic function of p110, prompting the phosphorylation of phosphatidylinositol 4,5-bisphosphate (PI(4,5)P_2_) to phosphatidylinositol 3,4,5-trisphosphate (PI(3,4,5)P_3_). PI(3,4,5)P_3_, a prominent second messenger in cell signalling, accumulates at the cell membrane attracting molecules with pleckstrin-homology (PH) domains, primarily phosphoinositide-dependent protein kinase 1 (PDK1) and the AKT serine/threonine kinase family (protein kinase B/PKB). PDK1 partially activates AKT at threonine 308 through phosphorylation, with full activation enabled by serine 473 phosphorylation via the mammalian target of rapamycin (mTOR) complex 2 (mTORC2). AKT targets various proteins in order to alter major signalling pathways within the cell. These include prosurvival pathways (BIM, BAX, BAD, BCL-2, MDM2 and FoxO1), cell cycle progression and glucose metabolism (p27, GSK3 and AS160), and cellular proliferation and protein synthesis (TSC2). Another prominent effector targeted downstream in the PI3K pathway is mTOR complex 1 (mTORC1), which has been shown to regulate cellular growth and metabolism, amongst other processes. In particular, mTORC1 targets ribosomal protein S6 kinase (S6K) and eukaryotic initiation factor 4E-binding protein 1 (4E-BP1), with the former directly effecting eukaryotic initiation factor 4E (eIF4E) and subsequent translation of cell cycle regulators.

**Figure 2 cancers-13-01538-f002:**
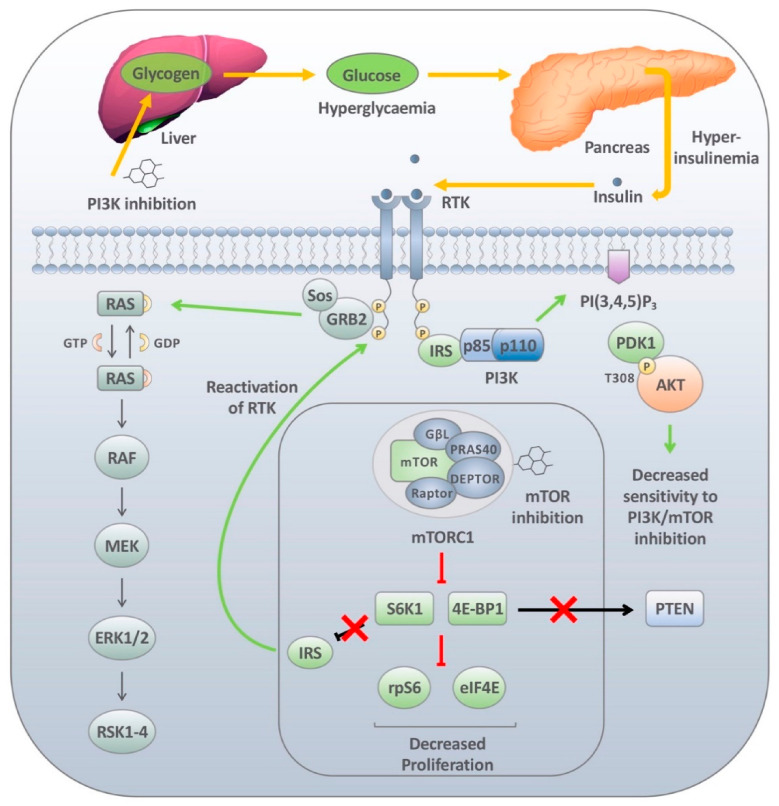
Insulin-mediated feedback loops following PI3K/mTOR inhibition. Following treatment with PI3K inhibitors, the liver breaks down stored glycogen releasing glucose into the bloodstream. The increased levels of glucose (hyperglycaemia) are detected by the pancreas, and in an effort to overcome these high levels of glucose, large amounts of insulin are released (hyperinsulinemia). This substantial release of insulin is sufficient to partially reactivate the insulin receptor which re-instates both IRS and GBR2 activity. What results is an increase in both PI3K and MAPK pathway activation, limiting the therapeutic effects of PI3K inhibitors. mTOR1 inhibitors, such as rapamycin, block downstream translation by downregulating S6K1 and 4E-BP1. In turn, this de-represses the S6K1 substrate IRS1, which acts as an intermediary between insulin receptor and the PI3K complex. The recruitment of PI3K to the active receptor enhances both MAPK and downstream PI3K signalling. Thus, limiting the overall sensitivity of mTOR inhibition in these tumours. Downregulation of mTOR activity either through AKT inhibition or direct mTOR inhibition blocks 4E-BP1-mediated translation of PTEN. This enhances the pool of PIP3 in cells resulting in the sustained activation of AKT.

**Figure 3 cancers-13-01538-f003:**
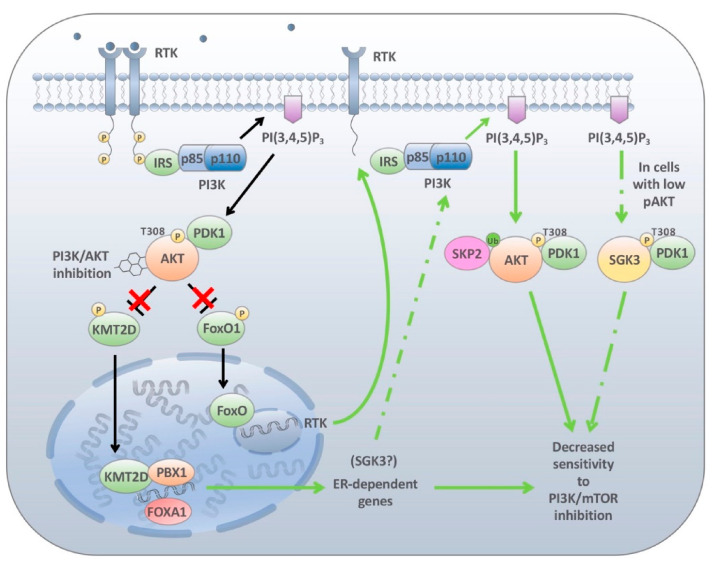
Adaptive and epigenetic-driven mechanisms of resistance to PI3K/AKT inhibition. A number of adaptive mechanisms regulating PI3K inhibitor sensitivity have been described. The most notable of which is the upregulation receptor tyrosine kinases (RTKs) following PI3K/AKT inhibition. Members of the FOXO family of transcription factors are direct substrates of AKT with phosphorylation limiting their nuclear localisation. Upon AKT downregulation FOXO is drawn into the nucleus where it is recruited to binding sites which upregulate a number of receptor tyrosine kinases including HER2 and HER3. This upregulation of these RTKs results in the enhanced activity of both MAPK and PI3K signalling pathways. Similarly, in ER+ breast cancers AKT phosphorylates the methyl-transferase KMT2D inhibiting the methyl-transferase activity of the enzyme. Upon AKT inhibition KMT2D activity is restored priming the recruitment of the transcription factors FOXA1, PBX1 and ER onto ER binding sites enhancing ER-dependent gene transcription and limiting the PI3K therapeutic effects. SGK3 is an ER transcriptional target. PIK3CA mutant tumours displaying low levels of pAKT can circumvent this by activating SGK3 and downstream mTOR signalling limiting the sensitivity of PI3K inhibitors.

**Table 1 cancers-13-01538-t001:** Inhibitors of the PI3K signalling pathway. Targeted therapies made specifically for the inhibition of elements of the PI3K signalling pathway. Although there are various targets and specificities for these therapies, only a few have been successful in receiving FDA approval for their effectiveness and patient response.

Target	Drug	Cancer Targets	FDA Status
Pan-PI3K Inhibitors	Copanlisib (BAY 80-6946)	Refractory follicular lymphoma (FL)	Approved
	Duvelisib (IPI-145)	Refractory follicular lymphoma (FL); refractory chronic lymphocytic leukaemia (CLL); small lymphocytic lymphoma (SLL); refractory follicular B-cell non-Hodgkin lymphoma (NHL)	Approved
	Buparlisib (BKM120)		Discontinued
	Pictilisib (GDC-0941)	Under clinical development in breast cancer	Under clinical development
Isoform-Specific PI3K Inhibitors	Alpelisib (BYL719)	Hormone receptor-positive/HER2-negative (HR+/HER2-) PIK3CA mutant breast cancer in combination with fulvestrant	Approved
	Idelalisib (CAL101)	Second-line treatment for patients with Chronic lymphocytic leukaemia (CLL) in combination with rituximab; follicular B-cell non-Hodgkin lymphoma (FL) and relapsed small lymphocytic lymphoma (SLL), both in patients who have received at least two prior systemic therapies.	Approved
	Serabelisib (INK1117/TAK-117)	Under clinical development for various tumours including breast, and endometrial cancer.	Active but not recruiting
AKT Inhibitors	MK2206	Under clinical development for PIK3CA and/or PTEN mutant breast cancer; non-small cell lung cancer, and ovarian cancers.	Under clinical development
	TAS-117		Under clinical development
	Capivasertib (AZD5363)	Under clinical development for patients with AKT E17K mutations	Under Clinical development
mTOR Inhibitors	Everolimus (RAD001)	Advanced renal cell carcinoma; hormone receptor-positive/HER2-negative (HR+/HER2-) breast cancer; gastrointestinal/lung neuroendocrine tumours (NET)	Approved
	Temsirolimus (CCI-779)	Advanced-stage renal cell carcinoma	Approved
	Torkinib (PP242)		Discontinued
	Sapanisertib (MLN0128)	Under clinical development for multiple solid tumours	Under clinical development
	Vistusertib (AZD2014)	Under clinical development for multiple solid tumours	Under clinical development
	Nab-sirolimus (ABI-009)	Perivascular epithelioid cell neoplasms (PEComa)	Approved
Dual PI3K/mTOR Inhibitors	Dactolisib (BEZ235)		Discontinued
	Apitolisib (GDC-0980)	Under clinical development for prostate cancer.	Under clinical development
	Vortalisib (XL765)		Discontinued
	Gedatolisib (PF-05212384)	Under clinical development for breast cancer	Under clinical development

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
