# Peer review of "Mechanisms of Resistance to PI3K Inhibitors in Cancer: Adaptive Responses, Drug Tolerance and Cellular Plasticity"

_cancers, 2021, doi:10.3390/cancers13071538_

Round 1

Reviewer 1 Report

This manuscript by Wright et al. provided an up-to-date overview of the current clinical status of the PI3K/AKT/mTOR pathway inhibitors, and a comprehensive review of the current understanding of the resistance mechanisms to these inhibitors. It will be of significant value to the scientific community in the field, and should be worth publishing in Cancers. There are several errors and issues, however, should be corrected and addressed before it’s accepted for publication.

Major point:

  • In Figure 1 and 2, PRAS is included in mTORC2 but not mTORC1. I believe PRAS40 is associated with mTORC1, while PRR5 is associated with mTORC2.

Specific points:

  • There are quite some grammatic, spelling and formatting errors throughout the text, which need to be corrected. These are annotated in the attached pdf file.

Author Response

Cancers-1115575

Mechanisms of resistance to PI3K inhibitors in cancer: Adaptive responses, drug tolerance, and cellular plasticity

Response to reviewer’s comments:

We thank the reviewers for the thoughtful and thorough review of the manuscript. We have revised the manuscript according to the reviewers’ suggestions and have addressed all the concerns brought forward by the reviewers.

Reviewer#1: ” It will be of significant value to the scientific community in the field, and should be worth publishing in Cancers.”

We thank the reviewer for regarding this review as a significant value to the scientific community.

Reviewer#1: In Figure 1 and 2, PRAS is included in mTORC2 but not mTORC1. I believe PRAS40 is associated with mTORC1, while PRR5 is associated with mTORC2. There are quite some grammatic, spelling and formatting errors throughout the text, which need to be corrected. These are annotated in the attached pdf file.

We thank the reviewer for their thorough revision of our manuscript. We have now edited figure 1 and 2 and corrected all of the suggestions in the main body of the text.

Reviewer#2: The manuscript suffers from a low degree of originality. I advise the authors to present and reference major other review works that present similar information and to highlight their own contribution compared to existing ones. Here is a list of very similar review works that should be referenced and discussed.

  1. Zhang, M.; Jang, H.; Nussinov, R. PI3K inhibitors: review and new strategies. Chem. Sci. 2020, 11, 5855–5865.
  2. Yang, J., Nie, J., Ma, X. et al. Targeting PI3K in cancer: mechanisms and advances in clinical trials. Mol Cancer 18, 26 (2019) – this work is referenced, but it should be presented as a similar review work.
  3. Nitulescu, G. M.; Van De Venter, M.; Nitulescu, G.; Ungurianu, A.; Juzenas, P.; Peng, Q.; Olaru, O. T.; Gradinaru, D.; Tsatsakis, A.; Tsoukalas, D.; Spandidos, D. A.; Margina, D. The Akt pathway in oncology therapy and beyond (Review). Int. J. Oncol. 2018, 53, 2319–2331.
  4. Xie, J.; Wang, X.; Proud, C. G. mTOR inhibitors in cancer therapy. F1000Research 2016, 5, F1000 Faculty Rev-2078.

We thank the reviewer for their reading of this manuscript. We have reread the reviews highlighted by the reviewer and referenced in the text the areas that were discussed in these reviews. However, in general we do not feel that there is a significant overlap between any of these reviews and this work which focuses on adaptive responses to PI3K pathway inhibitors. Specifically, the Zhang et al. review focuses on PIK3CA structural confirmations and the effect of oncogenic mutations within the PI3k structure. Furthermore, it beautifully highlights the selectivity of alpelisib and the binding of pi3k inhibitor to the catalytic pocket. In the Yang et al. review the authors have done an excellent job of highlighting all of the present ongoing clinical trials but the authors do not discuss any of the mechanisms of adaptive response to PI3K inhibition (the basis of our review) except for JAK signalling. The Nitulescu et al. ardently reviews PI3K signalling cascade but their review primarily focuses on AKT is various disease states. The review by Chris Proud has some overlap with the current rapalogs but this review does not discuss in any detail the mechanisms of resistance to these inhibitors.

Reviewer#2: The paper also needs a more critical approach. It should be more than a simple collection of data from various sources, but a critical analyze of them.

We thank the reviewer for their comment. We believe we have highlighted and analysed all of the seminal papers regarding this topic including our own. We believe in doing this we have written the most comprehensive and critical review on this topic to date.

Reviewer#2 Declare the source for the figure 1. It is original or an adapted version? Add the proper references.

We thank the reviewer for their comment. The figure is an original. However, we note that a number of reviews have been written on PI3K signalling highlighting the same downstream signalling cascades as we have.

Reviewer#2: The authors should focus more on PI3K inhibitors, and less on mTOR or Akt inhibitors. In my opinion these sections could be removed completely.

We thank the reviewer for their comment. We respectfully disagree. We argue that, due to the intrinsic feedback loops that exist in PI3K signalling it is likely that targeting multiple nodes of the pathway will be required to effectively downregulate PI3K/mTOR leading to increase in overall response rates. Therefore with the reviewers permission we would like to keep these sections.

Reviewer#2: In figure 2 are presented some drug examples. I advise the authors to delete the table from this figure. It should be presented as a standalone table with only PI3K inhibitors. It should also detail the clinical assays that are conducting on those inhibitors, and in case of the FDA approved drugs they should add the approved indications. What type of cancers can be treated with these drugs? I think the authors should also check and present the EMA approval status. I think the number of examples could be extended, as there are other compounds tested in several clinical trials.

We agree. We have now edited the figure/table in line with the reviewers suggestions.

Reviewer#2: The caption of Figure 3 should be similar to that of Figure 1 in explaining the key factors of the pathway.

This has now been edited in accordance with the reviewers comments.

Reviewer#3: In Figure 1 and 2, PRAS40 should be present in mTORC1 complex and not in mTORC2 and this is in line with the description in the text of mTORC1 regulation by PRAS40.

We thank the reviewer for editing this manuscript. In regards to this comment please see above.

Reviewer#3: In page 5, in the section where the authors mention the discovery of LY294002, they should also mention that this inhibitor is not that selective and can inhibit in addition of PI3K, mTOR and DNA-PK (doi: 1042/BJ20061489).

This has now been added.

Reviewer#3: In Figure 3, it would be good to include MAPK in the scheme as this is discussed in the text and in the legend (and Ref 44).

This has now been included.

Reviewer#3:General comments: can the authors include the side effects of the mTOR inhibitors , Rapalog? Etc…

We thank the reviewer for noticing this error on our part and have now included the side effects to rapalogs.

Reviewer#3:It would be good to add a comment on the following Refs ( (DOI 10.1016/j.ccr.2009.04.012 and DOI 10.15252/embj.201693929 ) reporting that some of PIK3CA mutant breast tumour cell lines show low levels of AKT activity and instead rely on SGK3-mediated signalling. Importantly, prolonged treatment with class I PI3K or AKT inhibitors induces a VPS34-dependent SGK3 signalling as an adaptive resistance response. This highlights the importance of considering intermittent rather than persistent drug treatment for cancer therapy.It would be good to mention the rebound effect of isoform-specific PI3K inhibitors thus limiting the use of isoform-selective PI3K inhibitors in certain cancers. Two seminal studies ( http://dx.doi.org/10.1016/j.ccell.2014.11.007 and http://dx.doi.org/10.1016/j.ccell.2014.11.008) identified and characterized several mechanisms that limit the ability of isoform-specific PI3K inhibitors when used as monotherapy to inhibit PI3K/Akt pathway and antitumor activity.

We thank the reviewer for their excellent suggestions. The following text has now been included.

 The mechanism of AKT reactivation in these above mentioned models is mediated by PI3K p110a. However, in PTEN null tumours PI3K p110b is the primary PI3K isoform driving PI3K signalling[164]. This discrepancy in PI3K activation through various cellular contexts highlights the many altered signalling routes that can lead PI3K mediated tumour proliferation. Moreover, it reasons that because these mutations affect various nodes of the same signalling cascade isoform specific PI3K inhibitors are unlikely to function in all genetic contexts. This was demonstrated in two seminal papers by Costa et al. and Schwartz et al [165,166]. Costa et al. demonstrated that inhibition of HER2-amplified cell lines with alpelisib resulted in a rebound of PIP3 levels after 6 hours [165]. This rebound was dependent upon p110b. Interestingly, in PIK3CA mutant tumours this rebound was not observed. Likewise, Schwartz et al. demonstrated that treatment of PTEN null cells with the p110b inhibitor AZD8186 significantly decreased PI3K signalling and tumour cell growth. In these models selective  p110a inhibition had no effect as these tumours are solely dependent upon p110b signalling. However, AKT/mTOR downregulation in these models was transient because downregulation of mTOR led to derepression of FOXO and subsequent RTK transcription leading to p110a mediated AKT signalling.  In both studies targeted inhibition with isoform specific PI3K inhibitors (p110a or p110b) triggered AKT/mTOR signalling, through the reactivation of the other PI3K isoform. Importantly, this reciprocal activation was annulled through the concomitant inhibition of p110a and p110b, resulting in greater antitumour activity compared to either inhibitor alone.

More recently, it has been demonstrated that as part of a negative feedback loop mTOR regulates PTEN translation through 4E-BP1[167]. The overall effect of this is to limit excessive PI3K signalling through the targeted regulation of PIP3 levels in the cells. Conversely, downregulation of PI3K signalling blocks mTOR phosphorylation of 4E-BP1 inhibiting PTEN translation, resulting in a rebound of AKT phosphorylation as soon as 2-4 hours post-treatment. A number of interesting conclusions were derived from these studies. The first was that PTEN downregulation was predominantly observed following PI3K inhibition with p110a, but not p110b inhibitors. Even though treatment with the p110b inhibitor AZD8186 effectively downregulated AKT signalling and a similar rebound in AKT signalling was observed at earlier time points. Critically, no downregulation of mTOR targets was observed upon treatment with AZD8186 at these early time points which likely explains the lack of PTEN downregulation. Nevertheless, it will be interesting to see if the AZD8186 mediated AKT rebound is dependent upon FOXO downregulation as determined by Schwartz et al. or if alternate rebound mechanisms exist. Importantly, these results also give credence to earlier observations by Costa et al. and may suggest the mechanism of action for the increase in PIP3 levels following alpelisib treatment in this previous work. Furthermore, this work confirms the observations in other studies, which advocate that variations in oncogenic mutations in the PI3K pathway significantly alters the levels of mTOR activity. It has been previously noted that cell lines harbouring PIK3CA mutations display lower levels of AKT phosphorylation compared to cell lines with PTEN loss [168]. In addition, cell lines with PTEN loss and activating mutations in PIK3CA do not confer the same level of AKT/mTOR signalling as cells with PTEN loss and coexistant PI3K activation mediated by upstream regulators (ex. RTKs) [167]. This is further confounded by the fact that cell lines with oncogenic helical mutations in PIK3CA display no demonstrable activation of AKT, even though they display equivalent levels of PDK1 expression and activation[168]. To circumvent this deficiency, PIK3CA mutant cell lines activate downstream mTOR signalling through the serum and glucocorticoid regulated kinase 3 (SGK3). SGK3 shares a 50% identity with the catalytic domain of AKT and can phosphorylate a number of AKT substrates including TSC2, promoting the activation of mTORC1. Downregulation of SGK3 by both genetic and chemical means significantly diminishes the viability of PIK3CA mutant cancer cells highlighting the importance of this signalling node in tumours with low AKT activation [168,169]. Although, SGK3 mediated signalling is not rapidly activated through a compensatory mechanisms following PI3K pathway inhibition, prolonged AKT inhibition does increase SGK3 mRNA levels over a number days[169]. The precise mechanism of SGK3 induction remains unknown; however, SGK3 has been reported to be an oestrogen transcriptional target in ER+ breast cancer suggesting that low levels changes in oestrogen signalling maybe sufficient to drive downstream mTOR activation, limiting the use of single agent PI3K pathway inhibitors (see below) [170].

Reviewer#3:Page3 : “AKT simultaneously phosphorylates the mTOR1 inhibitory subunit …” should be read “AKT simultaneously phosphorylates the mTORC1 inhibitory subunit proline-rich A. Page 12 , the reference [Mayer, 2014. #939] needs to be correclty edited.Page 14: “…Lin and colleagues demonstrated that FOXO can upregulated the mTORC2 component RICTOR resulting…” should read “…colleagues demonstrated that FOXO can upregulate the mTORC2 component RICTOR resulting…”

These suggestions have now been edited accordingly.

Reviewer 2 Report

The manuscript cancers-1115575, Mechanisms of resistance to PI3K inhibitors in cancer: Adaptive responses, drug tolerance, and cellular plasticity, presents a valuable review effort, but I consider it still needs important modifications in order to be published.

The manuscript suffers from a low degree of originality. I advise the authors to present and reference major other review works that present similar information and to highlight their own contribution compared to existing ones. Here is a list of very similar review works that should be referenced and discussed.

  1. Zhang, M.; Jang, H.; Nussinov, R. PI3K inhibitors: review and new strategies. Chem. Sci. 2020, 11, 5855–5865.
  2. Yang, J., Nie, J., Ma, X. et al. Targeting PI3K in cancer: mechanisms and advances in clinical trials. Mol Cancer 18, 26 (2019) – this work is referenced, but it should be presented as a similar review work.
  3. Nitulescu, G. M.; Van De Venter, M.; Nitulescu, G.; Ungurianu, A.; Juzenas, P.; Peng, Q.; Olaru, O. T.; Gradinaru, D.; Tsatsakis, A.; Tsoukalas, D.; Spandidos, D. A.; Margina, D. The Akt pathway in oncology therapy and beyond (Review). Int. J. Oncol. 2018, 53, 2319–2331.
  4. Xie, J.; Wang, X.; Proud, C. G. mTOR inhibitors in cancer therapy. F1000Research 2016, 5, F1000 Faculty Rev-2078.

The paper also needs a more critical approach. It should be more than a simple collection of data from various sources, but a critical analyze of them.

Declare the source for the figure 1. It is original or an adapted version? Add the proper references.

The authors should focus more on PI3K inhibitors, and less on mTOR or Akt inhibitors. In my opinion these sections could be removed completely.

In figure 2 are presented some drug examples. I advise the authors to delete the table from this figure. It should be presented as a standalone table with only PI3K inhibitors. It should also detail the clinical assays that are conducting on those inhibitors, and in case of the FDA approved drugs they should add the approved indications. What type of cancers can be treated with these drugs? I think the authors should also check and present the EMA approval status. I think the number of examples could be extended, as there are other compounds tested in several clinical trials.

The caption of Figure 3 should be similar to that of Figure 1 in explaining the key factors of the pathway.

The manuscript has multiple minor problems that are difficult to point out as the pdf file lacks the number for each line of text as it should have had.

Author Response

(The authors gave the same response as above.)

Reviewer 3 Report

In this review, the authors described well the PI3K pathway and the challenges of targeting this pathway for cancer therapy. However, in my view, there are few important studies that are missing and would need to be referenced to describe some aspects of the mechanism of resistance.

Therefore, few points need to be addressed before this MS could be considered for publication in Cancers.

Major points:

  • In Figure 1 and 2, PRAS40 should be present in mTORC1 complex and not in mTORC2 and this is in line with the description in the text of mTORC1 regulation by PRAS40.
  • In page 5, in the section where the authors mention the discovery of LY294002, they should also mention that this inhibitor is not that selective and can inhibit in addition of PI3K, mTOR and DNA-PK (doi: 1042/BJ20061489).
  • In Figure 3, it would be good to include MAPK in the scheme as this is discussed in the text and in the legend (and Ref 44).
  • General comments: can the authors include the side effects of the mTOR inhibitors , Rapalog? Etc…
  • It would be good to add a comment on the following Refs ( (DOI 10.1016/j.ccr.2009.04.012 and DOI 10.15252/embj.201693929 ) reporting that some of PIK3CA mutant breast tumour cell lines show low levels of AKT activity and instead rely on SGK3-mediated signalling. Importantly, prolonged treatment with class I PI3K or AKT inhibitors induces a VPS34-dependent SGK3 signalling as an adaptive resistance response. This highlights the importance of considering intermittent rather than persistent drug treatment for cancer therapy.
  • It would be good to mention the rebound effect of isoform-specific PI3K inhibitors thus limiting the use of isoform-selective PI3K inhibitors in certain cancers. Two seminal studies ( http://dx.doi.org/10.1016/j.ccell.2014.11.007 and http://dx.doi.org/10.1016/j.ccell.2014.11.008) identified and characterized several mechanisms that limit the ability of isoform-specific PI3K inhibitors when used as monotherapy to inhibit PI3K/Akt pathway and antitumor activity.

Minor points:

  • Page3 : “AKT simultaneously phosphorylates the mTOR1 inhibitory subunit …” should be read “AKT simultaneously phosphorylates the mTORC1 inhibitory subunit proline-rich A
  • Page 12 , the reference [Mayer, 2014. #939] needs to be correclty edited
  • Page 14: “…Lin and colleagues demonstrated that FOXO can upregulated the mTORC2 component RICTOR resulting…” should read “…colleagues demonstrated that FOXO can upregulate the mTORC2 component RICTOR resulting…”

Author Response

(The authors gave the same response as above.)

Round 2

Reviewer 2 Report

The authors made several significant modifications to the paper and thus improved its quality. Even if the authors haven't followed all my suggestion, I accept their decision to keep the  discussion on connecting pathways. I think the manuscript can be accepted for publishing.